# Role of Hepatocyte Senescence in the Activation of Hepatic Stellate Cells and Liver Fibrosis Progression

**DOI:** 10.3390/cells11142221

**Published:** 2022-07-17

**Authors:** Pramudi Wijayasiri, Stuart Astbury, Philip Kaye, Fiona Oakley, Graeme J. Alexander, Timothy J. Kendall, Aloysious D. Aravinthan

**Affiliations:** 1NIHR Nottingham Biomedical Research Centre, Nottingham University Hospitals NHS Trust and University of Nottingham, Nottingham NG7 2UH, UK; pramudi.wijayasiri@nottingham.ac.uk (P.W.); stuart.astbury@nottingham.ac.uk (S.A.); 2Nottingham Digestive Diseases Centre, Translational Medical Sciences, School of Medicine, University of Nottingham, Nottingham NG7 2UH, UK; 3Department of Pathology, Nottingham University Hospitals NHS Trust, Nottingham NG7 2UH, UK; philip.kaye@nuh.nhs.uk; 4Bioscience Institute, Faculty of Medical Sciences, Newcastle University, Newcastle upon Tyne NE2 4HH, UK; fiona.oakley@newcastle.ac.uk; 5UCL Institute for Liver & Digestive Health, Division of Medicine, Royal Free Campus, London NW3 2PF, UK; g.alexander@ucl.ac.uk; 6Institute for Regeneration and Repair, Centre for Inflammation Research, University of Edinburgh, Edinburgh EH16 4TJ, UK; tim.kendall@ed.ac.uk

**Keywords:** hepatocyte senescence, senescence-associated secretory phenotype (SASP) factors, hepatic stellate cells, smooth muscle alpha-actin (αSMA)

## Abstract

Hepatocyte senescence is associated with liver fibrosis. However, the possibility of a direct, causal relation between hepatocyte senescence and hepatic stellate cell (HSC) activation was the subject of this study. Liver biopsy specimens obtained from 50 patients with non-alcoholic fatty liver disease and a spectrum of liver fibrosis stages were stained for p16, αSMA, and picrosirius red (PSR). Primary human HSCs were cultured in conditioned media derived from senescent or control HepG2 cells. Expression of inflammatory and fibrogenic genes in HSCs cultured in conditioned media were studied using RT-PCR. ELISAs were undertaken to measure factors known to activate HSCs in the conditioned media from senescent and control HepG2 cells and serum samples from healthy volunteers or patients with biopsy-proven cirrhosis. There was a strong association between proportion of senescent hepatocytes and hepatic stellate cell activation. Both proportion of hepatocyte senescence and hepatic stellate cell activation were closely associated with fibrosis stage. Inflammatory and fibrogenic genes were up-regulated significantly in HSCs cultured in conditioned media from senescent HepG2 cells compared with control HepG2 cells. PDGF levels were significantly higher in the conditioned media from senescent hepatocytes than control HepG2-conditioned media, and in serum samples from patients with cirrhosis than healthy volunteers. In conclusion, this ‘proof of concept’ study revealed activation of human HSCs by media from senescent HepG2 cells, indicating direct involvement of factors secreted by senescent hepatocytes in liver fibrosis.

## 1. Introduction

Senescent cells no longer replicate, but retain some metabolic activity. Cells become senescent to avoid impending DNA damage being passed on to daughter cells. Senescent cells are morphologically and metabolically distinct from their replicating counterparts [1]. There are two broad paths to cell senescence. Replicative senescence, where cells becoming senescent after a fixed number of divisions, was discovered in yeast in 1959 [2]. Before replicative senescence develops, several sub-lethal cell stressors such as oxidative stress, ionising radiation, osmotic stress, or hypoxia can trigger senescence prematurely, known as stress-induced premature senescence [3].

The metabolic activity of senescent cells is governed by the senescence-associated secretory phenotype (SASP), which consists of a vast array of cytokines, chemokines, proteases, and growth factors [4,5].

Accumulation of senescent hepatocytes is well described in chronic liver disease, irrespective of aetiology [6,7,8,9,10]. Most studies demonstrated a strong positive association between the proportion of senescent hepatocytes and liver fibrosis stage [6,7,8,10]. However, it is not clear whether progression of liver fibrosis is a consequence of the accumulation of senescent hepatocytes or whether accumulation of senescent hepatocytes and liver fibrosis progression are unrelated consequences of liver injury. The purpose of this study was to investigate whether senescent hepatocytes play a role in liver fibrosis.

Hepatic stellate cells (HSCs) are resident perisinusoidal cells in the subendothelial space between hepatocytes and sinusoidal endothelial cells [11]. Their location within the perisinusoidal space (space of Disse) and their multiple processes, which extend around the perisinusoidal space, allow interaction with hepatocytes and endothelial cells [11]. HSCs play an important role in the liver’s response to injury [11]. They are activated by oxidative stress, apoptotic fragments, and a variety of cytokines including platelet-derived growth factor (PDGF), transforming growth factor beta (TGFβ), epidermal growth factor (EGF), Amphiregulin (AREG), and Vascular endothelial growth factor (VEGF) [11,12,13]. Activation of HSCs causes major morphological change and evolution into myofibroblast-like cells. Induction of alpha-smooth muscle actin (α-SMA) is the single most reliable marker of HSC activation [11].

## 2. Materials and Methods

### 2.1. Liver Biopsy Specimens and Histological Quantification

All liver biopsy specimens were obtained in accordance with local research and ethics committee guidelines. Formalin-fixed, paraffin-embedded sections from 50 liver biopsies with confirmed non-alcoholic fatty liver disease (NAFLD) across the spectrum of fibrosis were obtained.

Sections were stained for αSMA (αSMA-DAKO-GA611 concentration 1:200; Heat Induced Epitope Retrieval (HIER) in Tris-based CC1 buffer pH 8–8.5 at 95 °C for 64 min), a marker of HSC activation [11], and p16 (Roche-concentration 1:200; heat-induced epitope retrieval (HIER) in Tris-based CC1 buffer pH 8–8.5 at 95 °C for 64 min), an established marker of permanent cell cycle arrest and therefore cell senescence [14], to assess the association between proportion of senescent hepatocytes, HSC activation, and liver fibrosis.

Additional sections were stained with picrosirius red (PSR) and scored using a 5-tier staging system (F0-no fibrosis; F1—mild fibrosis; F2—moderate fibrosis F3—severe fibrosis; and F4—cirrhosis) by an independent liver histopathologist (P.K.) blinded to αSMA expression results.

αSMA expression was quantified in sections using immunohistochemistry by a second independent pathologist (T.J.K.) blinded to all other results. Whole-slide images were split using ndpisplit [15] into tiles of ×5 magnification before the application of a classifier identifying positive staining that had been trained using the machine learning WEKA plugin in FIJI [16,17], as described previously. This generates the total number and percentage of pixels classified as αSMA positive. Formal observer-independent image analysis to quantify αSMA was undertaken to remove any subjective interpretation.

The proportion of senescent hepatocytes was assessed quantitatively to ensure objectivity through identifying p16-positive hepatocytes using FIJI [17]. Three fields at ×20 magnification were randomly selected from each slide, images were deconvoluted using the H-DAB setting in FIJI, followed by thresholding on the blue channel and the analysing particles function to count p16-negative cells. p16-positive cells were counted manually and expressed as a percentage of total cells.

### 2.2. Induction of Cellular Senescence and Hepatic Stellate Cell Culture

Cellular senescence was induced and confirmed in the human liver cell line HepG2 (ATCC HB-8065) as described previously [18,19]. In brief, the HepG2 cells (ATCC HB-8065) were grown in Dulbecco’s Modified Eagle Medium containing 10% fetal calf serum and antibiotics (100 U/mL Penicillin, 100 µg/mL Streptomycin). Cells were seeded in 6-well plates at a density of 5 × 10^5^ per well and allowed to adhere overnight. Cells were then incubated with 0.5 mM H2O2 in culture media for 60 min to induce senescence (the optimal conditions for induction of senescence rather than apoptosis or cell death). Control HepG2 cells were incubated in culture media alone. After senescence was established, the HepG2cells were washed three times with PBS and incubated in fresh culture medium for 5 further days. Then the culture media from control or H2O2-treated (i.e., senescent) HepG2 cells was filtered with a 0.2 µm syringe filter to render cell-free conditioned media.

In brief, HepG2 senescence was confirmed by demonstration senescence β-Galactosidase expression (Cell Signaling Technology^TM^) and morphological changes under light microscopy, demonstration of p21 expression (Alexa Fluor^®^ 488-conjugated rabbit anti-p21; Cell Signaling Technology; concentration 1:50), and demonstration of senescence-associated heterochromatic foci (unconjugated rabbit anti-HMGI-C; Santa Cruz Biotechnology; concentration 1:500; FITC fluorochrome-conjugated donkey anti-rabbit secondary antibody) using immunofluorescence.

HSCs were isolated by sequential perfusion with collagenase (Roche) and pronase (Roche), followed by discontinuous density centrifugation in 11.5% Optiprep (Life Technologies, UK) from normal margins of adult male human livers following partial hepatectomy for metastatic cancer performed on consenting patients, as described previously [20]. Collection and use of human tissue was ethically approved by North East-Newcastle and North Tyneside 1 research committee (H10/H0906/41) and was conducted in Professor Fiona Oakley’s laboratory at Newcastle University. HSCs were seeded at a density of 3 × 10^5^ cells per well on 6-well plates and incubated overnight at 5% CO_2_ and 37 °C in DMEM plus L-glutamine, Na-pyruvate, and antibiotics supplemented with 10% foetal bovine serum. Following a wash with PBS, HSCs were incubated in conditioned media from senescent or control HepG2 cells mixed with standard fresh media (ratio 50:50) for 6 or 24 h. Freshly isolated HSC (day 0) were considered quiescent and only freshly isolated human HSCs were used in these experiments; transdifferentiation to fully activated HSC was only observed after continuous culture for 10 days or more.

HSC activation was measured by inflammatory and fibrogenic gene upregulation. RNA was extracted from HSCs cultured in conditioned culture media from senescent or control HepG2 cells using TRIZOL^®^ Reagent. RT-PCR was performed with SYBR Green JumpStart Taq ready mix (Sigma-Aldrich) according to the manufacturer’s instructions. The forward and reverse sequences of human primers used are summarised in Table 1. The annealing temperature was 55 °C for all primers used. Expression of all genes was normalised to Glyceraldehyde 3-phosphate dehydrogenase (GAPDH) levels.

### 2.3. Determination of Fibroblast Activating Factors in Culture Media and Serum Samples

Factors well-known to activate fibroblasts and HSCs (e.g., PDGF, TGFβ1, LTBP1, and AREG) were quantified using enzyme-linked immunosorbent assay (ELISA) in conditioned media from senescent or control HepG2 cells and in serum samples from 10 healthy volunteers and 10 patients with biopsy-proven NAFLD cirrhosis. This was ethically approved by Wales research committee 5 (14/WA/1234).

### 2.4. Statistical Analysis

All experiments were conducted in triplicate, unless stated otherwise. Statistical analysis was performed using GraphPad prism 9 and SPSS (version 27). *p* < 0.05 was considered significant. Two-tailed unpaired t-test was used to define differences between means of normally distributed data of equal variance. Mann–Whitney test was used for data that were not distributed normally. Comparison between multiple groups was conducted using 1-way ANOVA (Kruskal–Wallis test).

## 3. Results

### 3.1. Association between Hepatocyte Senescence, Activation of Hepatic Stellate Cells, and Liver Fibrosis

The demographic and clinical characteristics of 50 patients with NAFLD are summarised in Table 2; the median age was 57 years (IQR 48–67); 30 (60%) were males; 30 (60%) had at least one metabolic syndrome condition (type 2 diabetes mellitus, hypertension, dyslipidaemia or obesity). Forty-one patients (82%) had at least some degree of steatohepatitis.

There was a significant positive correlation between hepatocyte p16 expression and αSMA expression in human liver biopsies with NAFLD (linear regression *p* < 0.0001, R^2^ = 0.7244, Figure 1A). Similarly, there was a positive correlation between fibrosis stage and hepatocyte p16 expression (Kruskal–Wallis test *p* < 0.0001, Figure 1B) and αSMA expression (Kruskal–Wallis test *p* = 0.0001, Figure 1C). Areas of increased hepatocyte p16 expression were closely associated with increased HSC activation, as indicated by increased αSMA expression and thick fibrous bands (Figure 2) and areas of low p16 expression were closely associated with reduced SMA expression and less fibrous tissue (Figure 2).

### 3.2. Induction of Senescence in Hepg2 Cells

Incubation with 0.5 mM H_2_O_2_ in culture medium for 60 min induced cell senescence in >95% of the HepG2 cells at 5 days’ post-exposure. Senescence was confirmed by expression of senescence β-Galactosidase, characteristic morphological changes such as enlargement, flattening, and elongation in H_2_O_2_-treated HepG2 cells. More than 95% of H_2_O_2_-treated HepG2 cells expressed the cell cycle inhibitor, p21; >90% expressed senescence-associated heterochromatic foci, HMGI-C, confirming the induction of cell senescence (Figure 3).

### 3.3. Induction of Hepatc Stellate Cells

There was significant up-regulation of inflammatory genes including TNFα (*p* = 0.02) and IL-1β (*p* = 0.04), but not IL-6 (*p* = 0.72) in HSCs cultured in media derived from senescent HepG2 cells compared with control HepG2 cells. Similarly, a significant up-regulation of fibrogenic genes including SMA (*p* = 0.02), TIMP1 (*p* = 0.03) and ProCollagen (*p* = 0.048) was evident in HSCs cultured in media derived from senescent HepG2 cells compared with control HepG2 cells (Figure 4). In general, fibrotic gene expression was more marked in HSCs cultured in senescent HepG2 media for 24 h than for 6 h.

### 3.4. SASP Factors

Levels of PDGF were significantly higher in conditioned media from senescent HepG2 cells compared with media from control HepG2cells (Figure 5), but there were no differences in TGFβ1, LTBP1, and AREG levels. A similar pattern of significant elevation of PDGF levels was noted in serum samples from patients with cirrhosis compared to healthy volunteers (Figure 5).

## 4. Discussion

This study demonstrated a direct, causal association between hepatocyte senescence and liver fibrosis. HSCs are the major source of liver fibrosis and activation of HSCs is a key event in fibrogenesis [11]. HSC activation occurs in two phases: initiation and perpetuation [11,12]. During initiation, changes occur in gene expression which leads to the perpetuation phase, where HSCs become pro-inflammatory and fibrogenic. Previous studies demonstrated a strong positive association between the proportion of senescent hepatocytes and liver fibrosis stage, irrespective of aetiology, but did not address a causal association [6,8,9,10]. The current study demonstrated that senescent hepatocytes influence their microenvironment through secretion of senescence-associated secretory factors; incubation of HSCs in cell-free conditioned media derived from senescent HepG2 cells led to changes in their gene expression profile. Some of the genes responsible for a pro-inflammatory state and fibrogenesis were up-regulated, potentially indicating the activation of HSCs (initiation phase) by the conditioned media from senescent HepG2 cells.

HepG2 cells are frequently used as in vitro alternatives to primary human hepatocytes to study liver disease [21]. However, their use in understanding human chronic liver disease has been questioned. In this study, HepG2 cells were used to evaluate the role of senescent hepatocyte in liver fibrosis progression. The senescent HepG2 cell gene signature has been shown to be highly comparable to human chronic liver diseases such as non-alcoholic fatty liver disease, alcohol-related liver disease and hepatitis C related liver disease [18]. Further, senescent HepG2 cell models have also been used successfully to explore the pathophysiology of chronic liver disease [19,22,23]. The above detailed evidence validates and supports the use of senescent HepG2 cell models for the understanding of human chronic liver disease.

Senescent cells are known to exert influence on neighbouring cells and their microenvironment through a wide variety of factors known as the senescence-associated secretary phenotype (SASP). These factors promote inflammation, which aids clearance of cell debris and encourages tissue repair and remodelling [1,4,9,24,25]. Attraction and activation of fibroblasts to lay down fibrosis is a crucial step in limiting tissue damage and enabling wound healing. Unfortunately, the accumulation of senescent hepatocytes due to the continuous, chronic injury characteristic of chronic liver disease results in a greater number of HSCs being activated and, therefore fibrosis progression, cirrhosis, and its complications such as portal hypertension—an unintended adverse consequence of cellular senescence and SASP factors.

The most important question arising from this study is whether this new understanding of the role of senescent hepatocytes and SASP factors in liver fibrosis can be exploited for therapeutic advantage. Autophagy, which allows cells under stress to digest internal constituents to generate energy and metabolic precursors, is closely associated with senescence [26]. This pathway is coupled to the mammalian target of rapamycin (mTOR)-associated anabolic pathway, allowing protein degradation to feed raw materials directly into protein synthesis for the production of SASP factors. mTOR inhibitors have been shown downregulates SASP factors [27,28]. It would not be unreasonable to hypothesise that mTOR inhibitors such as sirolimus could be used as anti-fibrotic agents in chronic liver disease. This notion is supported by the profound in vitro anti-fibrotic effect of sirolimus, a potent mTOR inhibitor [29]. Further, sirolimus has been shown to reduce fibrosis, improve liver function, and improve survival in animal models, even after the development of cirrhosis [30,31]. In humans, a retrospective analysis of patients who underwent liver transplantation for hepatitis C showed a significant reduction in the rate of fibrosis progression and improved survival with sirolimus when it was used as an anti-rejection treatment [32].

Finally, this study demonstrated that αSMA could be used as an early biomarker for liver fibrosis. αSMA is considered the single most reliable marker of activated HSCs [11]. There was an increment in the expression of αSMA from fibrosis stage F0 (no fibrosis) to fibrosis stage F4 (cirrhosis) (Figure 1C). Further, the data also indicate that a 50% reduction in αSMA is likely to translate into a clinically meaningful reduction in liver fibrosis (i.e., one stage reduction in fibrosis) strengthening the notion that αSMA is an early biomarker of liver fibrosis. A reduction in activated HSCs, as indicated by a reduction in αSMA, has been shown to correlate closely with a reduction in fibrosis in previous studies. In an animal study [30], a 50% reduction in αSMA paired with a 35% reduction in liver fibrosis was observed with 14 days of sirolimus treatment. Similarly, another animal study [31] demonstrated an 85% reduction in αSMA and a subsequent 82.5% reduction in liver fibrosis with 14-days of sirolimus treatment. In another study, successful eradication of hepatitis C led to a 50% reduction in αSMA followed by significant improvement in liver fibrosis stage [33]. Taken together, these studies corroborate the use of αSMA as an early surrogate of liver fibrosis.

This study has its own strengths and weaknesses. The use of HepG2 cells, a transformed cell line could be considered a limitation of this study. However, previous studies show HepG2 senescence as a viable model to study the role of hepatocyte senescence in chronic liver disease [18,19,34]. Further, the lack of a universal biomarker for the unequivocal detection of senescent cells in vitro and in vivo [35] is an important limitation in the field of senescence. However, the use of p16, a marker of permanent cell cycle arrest and established cell senescence, is considered a unique and specific method of identifying senescent cells [36]. An important strength of this study is the potential demonstration of a causal link between cellular senescence and fibrosis progression in chronic disease. This finding is supported by a previous study, which demonstrated promotion of quiescent stromal fibroblasts activation by senescent cholangiocytes through PDGF, a SASP factor [37]. Together, these studies validate the notion that epithelial senescence promotes fibrosis progression in chronic diseases of multiple organs as seen in fibrotic pulmonary disease [38], and thus facilitate novel anti-fibrotic treatment options (as discussed above). Our previous study, that investigated changes in gene expression in senescent hepatocytes, indicated increased expression of fibrotic SASP factors such as TGFβ1, LTBP1, and AREG [18]. However, this does not seem to be translated into increased protein synthesis of some of the SASP factors that were investigated in the current study. Although the reason(s) behind this dissociation is not apparent, previous studies have also demonstrated such differential regulation of the production of SASP factors [27,28].

In this hypothesis-generating preliminary study, hepatocyte senescence plays a key role in liver fibrosis progression mediated by SASP factors, which could be exploited for therapeutic advantage.

## Figures and Tables

**Figure 1 cells-11-02221-f001:**
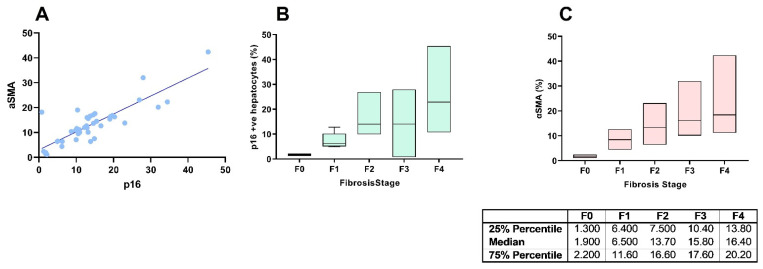
Associations between proportion of senescent hepatocyte (represented by positive p16 staining) and hepatic stellate cell activation (represented by αSMA expression) (**A**); proportion of senescent hepatocytes (represented by positive p16 staining) and liver fibrosis stage (**B**); hepatic stellate cell activation (represented by αSMA expression) and liver fibrosis stage (**C**).

**Figure 2 cells-11-02221-f002:**
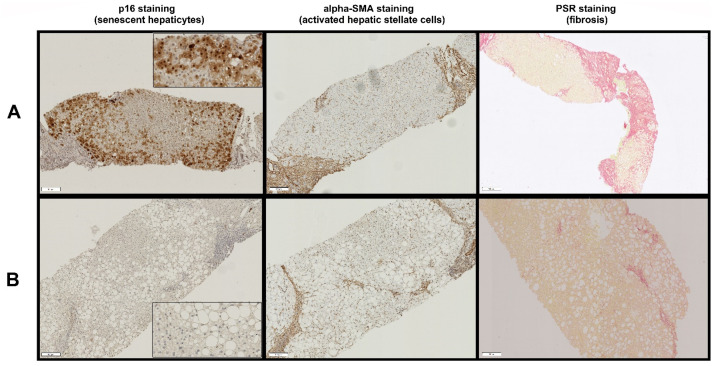
Association between p16 and αSMA expression and fibrous tissue using serial 10 µm sections cut from the same biopsy. High hepatocyte p16 expression was closely associated with increased hepatic stellate cell activation (shown by high αSMA expression) and thick fibrous tissue (**panel A**), and low p16 expression was closely associated with low αSMA expression and thin fibrous tissue (**panel B**). Magnification is at 10× with insets for p16 expression is shown at magnification 40×. Scale bar length is 100 µm at 10× magnification.

**Figure 3 cells-11-02221-f003:**
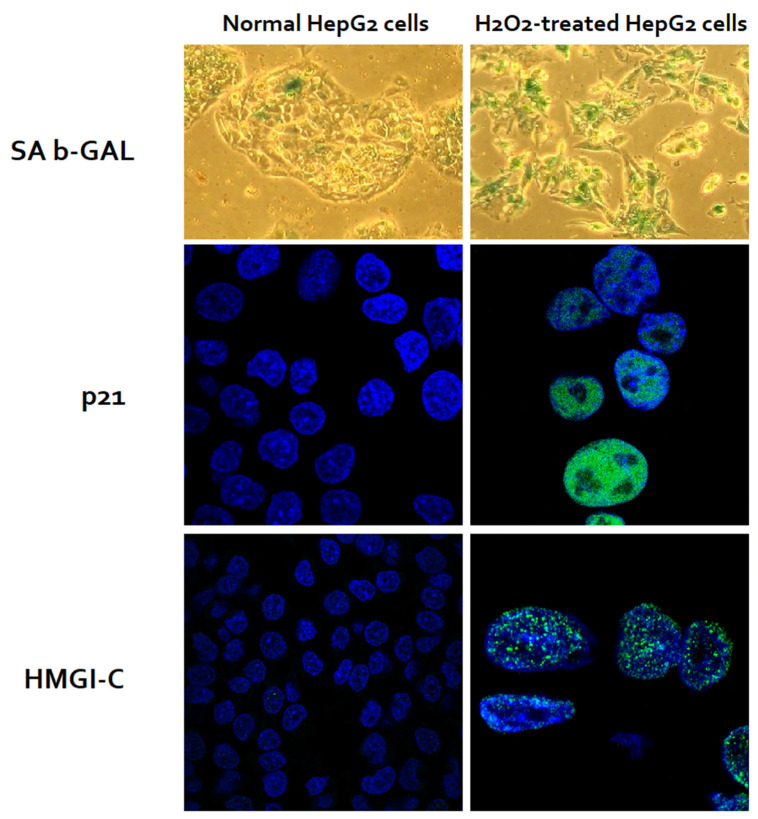
Induction of senescence I HepG2 cells through oxidative stress using H_2_O_2_. Of the cells, >95% of H_2_O_2_-treated HepG2 cells and <5% of untreated HepG2 cells expressed SA-β-GAL (green staining). H_2_O_2_-treated HepG2 cells were enlarged, and flattened with significantly larger nuclei. More than 95% of the H_2_O_2_-treated, but none of the untreated HepG2 cells, expressed p21 (a cell cycle inhibitor, green nuclear stain). Similarly, more than 90% of the H_2_O_2_-treated, but none of the untreated HepG2 cells, expressed HMGI-C (a senescence-associated heterochromatic foci, green nuclear stain). This figure is reproduced with modifications with permission from Aravinthan A et al., *Experimental Gerontology* (2014) [18].

**Figure 4 cells-11-02221-f004:**
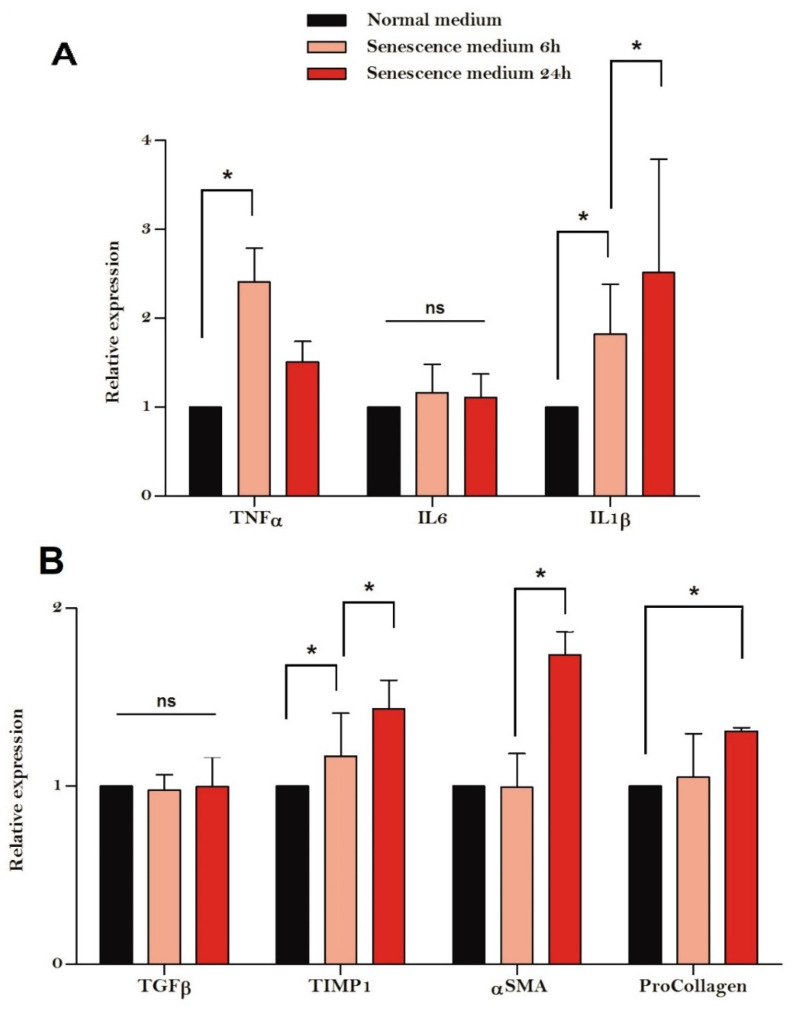
Inflammatory (**A**) and fibrotic (**B**) gene expression of hepatic stellate cells cultured in conditioned HepG2 media. Inflammatory genes such as Tumour necrosis factor alpha (TNFα) and Interleukin 1 beta (IL1β) but not Interleukin 6 (IL6) were up-regulated in hepatic stellate cells cultured in media from senescent HepG2 cells. Similarly, fibrotic genes such as smooth muscle actin alpha (αSMA), Tissue inhibitor of metalloproteinases 1 (TIMP1) and ProCollagen, but not Transforming growth factor beta (TGFβ), were up-regulated in hepatic stellate cells cultured in media from senescent HepG2 cells. Experiments were performed in triplicate and the results are expressed as means with SD. The asterisk (*) indicates *p* < 0.05 and ns indicates no statistical significance.

**Figure 5 cells-11-02221-f005:**
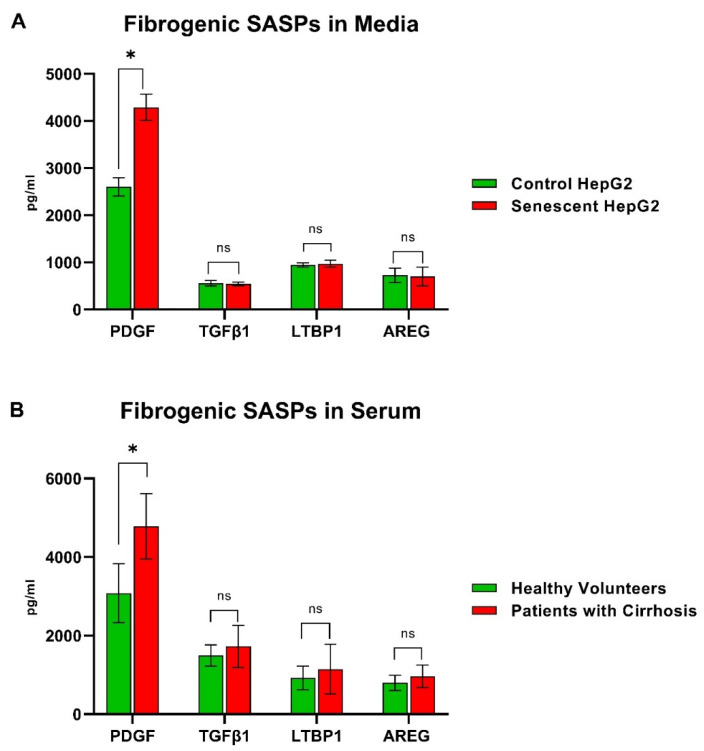
Levels of TGFβ1, LTBP1, AREG, and PDGF, potent hepatic stellate cell activators (fibrogenic senescence-associated secretory factors), in conditioned media from senescent HepG2 cells compared with media from the control HepG2cells (**A**), and in serum samples from patients with cirrhosis compared with serum samples from healthy volunteers (**B**). Experiments were performed in triplicate and the results are expressed as means with SD. The asterisk (*) indicates *p* < 0.05 and ns indicates no statistical significance.

**Table 1 cells-11-02221-t001:** Summary of forward (F) and reverse (R) sequences of human primers.

Primer		Sequence (5′ to 3′)
*GAPDH*	F	GAAGGTGAAGGTCGGAGTC
	R	GAAGATGGTGATGGGATTTC
*IL-6*	F	AAATTCGGTACATCCTCGACGG
	R	GGAAGGTTCAGGTTGTTTTCTGC
*TNFα*	F	AACCTCCTCTCTGCCATCAA
	R	CCAAAGTAGACCTGCCCAGA
*IL-1β*	F	TTCCATTTTGTTTGCCTTAT
	R	TTATGGGAAAGTCTCAAAAC
*TIMP1*	F	TTATCCATCCCCTGCAAACTG
	R	CAAGGTGACGGGACTGGAA
*αSMA*	F	GCGTGGCTATTCCTTCGTTACT
	R	CCGATGAAGGATGGCTGGAACA
*ProCollagen I*	F	CAAGAGGAAGGCCAAGTCGAG
	R	CGTTGTCGCAGACGCAGAT
*TGFβ1*	F	TGACAGCAGGGATAACACACT
	R	CGCACGCAGCAGTTCTTCTCC

Abbreviations: GAPDH Glyceraldehyde 3-phosphate dehydrogenase; IL-6 Interleukin 6; TNFα Tumour necrosis factor alpha; IL-1β Interleukin 1 beta; TIMP1 Tissue inhibitor of metalloproteinases 1; *αSMA* Smooth muscle actin alpha; *TGFβ* Transforming growth factor beta.

**Table 2 cells-11-02221-t002:** Demographic and clinical characteristics of patients with non-alcoholic fatty liver disease (NAFLD).

		All (n = 50) Median (IQR) or Number (%)
**Age at biopsy**		57 (48–67)
**Female sex**		20 (40%)
**Body Mass Index (kg/m** ^ **2** ^ **)**		36.8 (30.7–40.4)
**Metabolic Comorbidities**	**Type 2 DM**	26 (52%)
	**Hypertension**	20 (40%)
	**Dyslipidaemia**	27 (54%)
**Blood tests at biopsy**	**AST**	37 (31–56)
	**ALT**	39 (29–59)
	**ALP**	90 (73–117)
	**Bilirubin**	10 (8–13)
	**Albumin**	42 (40–44)
	**AST/ALT ratio**	1.0 (0.8–1.1)
	**Platelets**	255 (197–302)
	**PT**	11.0 (10.8–11.3)
	**HbA1c**	50 (39–63)
**Biopsy Steatosis**	**Mild (grade 1)**	5 (10%)
	**Moderate (grade 2)**	22 (44%)
	**Severe (grade 3)**	23 (46%)
	**Steatohepatitis**	41 (82%)
**Biopsy Fibrosis Stage**	**0 (no fibrosis)**	7 (14%)
	**1**	9 (18%)
	**2**	11 (22%)
	**3**	11 (22%)
	**4 (cirrhosis)**	12 (24%)

## Data Availability

Data of this study are available from the corresponding author upon reasonable request and an appropriate institutional collaboration agreement.

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
