# Peer review of "Role of Hepatocyte Senescence in the Activation of Hepatic Stellate Cells and Liver Fibrosis Progression"

_cells, 2022, doi:10.3390/cells11142221_

Round 1
Reviewer 1 Report
The authors have addressed most of the comments improving the formal correctness of the manuscript. Nevertheless, more substantial issues are left unchanged by this revision:
1) the discrepancy between the SAPS profile (TGFb, LTBP1 and AREG) in their own previous publication is not discussed
2) The extent of SAPS evaluation is very limited, since only PDGF, TGFb LTBP1 and AREG were evaluated, both in conditioned media and patients’ sera
3) I acknowledge that this work has the merit of using freshly isolated human HSC. However, mechanistic investigations on SAPS-induced HSC activation are, at best, very preliminary. In particular, there is no demonstration that PDGF released by senescent HepG2 promotes the activation of HSC. Indeed, HSC have been shown to express PDGFR only after several days in colture (Wong L, Yamasaki G, Johnson RJ, and Friedman SL (1994). Induction of β-platelet-derived growth factor receptor in rat hepatic lipocytes during cellular activation in vivo and in culture. J Clin Invest 94: 1563–1569.)
Moreover, quiescent freshly isolated rat HSC do not express PDGFR (Kinnman, N., Goria, O., Wendum, D. et al. Hepatic Stellate Cell Proliferation is an Early Platelet-Derived Growth Factor-Mediated Cellular Event in Rat Cholestatic Liver Injury. Lab Invest 81, 1709–1716 (2001). https://doi.org/10.1038/labinvest.3780384).
Since the isolation of HSC requires protease digestion, there is further concern that a significant amount of PDGFR may be available on the surface of freshly isolated HSC in within the time frame of the experiments conducted here.
Therefore, authors are required to convincingly show the expression of PDGFR on freshly isolated HSC. They should also verify that treatment of freshly isolated HSC with PDGF (for 6hs or 24hs) parallels the use of senescent-conditioned media on HSC activation.
4) The authors have assessed hepatocytes senescence in liver biopsies by p16 staining, but then used p21 in HepG2 in vitro model. Although both can be considered markers of senescence, the choice of either marker should be explained.
Minor:
Fig4. it is not clear for which of the treatment group (6hs or 24hs) the significance is reached. Please put the asterisk above the bar for which significance is reached
Author Response
The authors have addressed most of the comments improving the formal correctness of the manuscript. Nevertheless, more substantial issues are left unchanged by this revision:
- the discrepancy between the SAPS profile (TGFβ, LTBP1 and AREG) in their own previous publication is not discussed
As reviewer point out, our previous publication that explored gene expression of senescent hepatocytes demonstrated increased expression of PDGF, TGFβ1, LTBP1 and AREG in HepG2 senescence However, in the current study we have explored the protein expression and it seems that there is a dissociation between gene expression and protein expression in some of the SASP factors. Although the reason/s for this is not readily identifiable, pother studies have also shown similar dissociation between gene expression and protein expression in senescent cells. We have acknowledged this in the DISCUSSION section.
- The extent of SAPS evaluation is very limited, since only PDGF, TGFβ, LTBP1 and AREG were evaluated, both in conditioned media and patients’ sera
As the reviewer points out, we have not investigated the entire panel of SASP factors that could potentially be involved in the activation of hepatic stellate cells. This is a hypothesis-generating preliminary study and we only investigated the most common SASP factors that are known to activate fibroblasts and that were shown to be upregulated in our above-mentioned previous study(Aravinthan, A., et al., Exp Gerontol, 2014).
- I acknowledge that this work has the merit of using freshly isolated human HSC. However, mechanistic investigations on SAPS-induced HSC activation are, at best, very preliminary. In particular, there is no demonstration that PDGF released by senescent HepG2 promotes the activation of HSC. Indeed, HSC have been shown to express PDGFR only after several days in culture (Wong L, Yamasaki G, Johnson RJ, and Friedman SL (1994). Induction of β-platelet-derived growth factor receptor in rat hepatic lipocytes during cellular activation in vivo and in culture. J Clin Invest 94: 1563–1569.)
Moreover, quiescent freshly isolated rat HSC do not express PDGFR (Kinnman, N., Goria, O., Wendum, D. et al. Hepatic Stellate Cell Proliferation is an Early Platelet-Derived Growth Factor-Mediated Cellular Event in Rat Cholestatic Liver Injury. Lab Invest 81, 1709–1716 (2001). https://doi.org/10.1038/labinvest.3780384).
Since the isolation of HSC requires protease digestion, there is further concern that a significant amount of PDGFR may be available on the surface of freshly isolated HSC in within the time frame of the experiments conducted here.
Therefore, authors are required to convincingly show the expression of PDGFR on freshly isolated HSC. They should also verify that treatment of freshly isolated HSC with PDGF (for 6hs or 24hs) parallels the use of senescent-conditioned media on HSC activation.
As we mention in the manuscript, this is a hypothesis-generating preliminary study, that was set to investigate the potential causal link between hepatocyte senescence and liver fibrosis. The study was not aimed to investigate the role of PDGF in stellate cell activation. Further, we, as a group do not have the expertise to carry out an in-depth investigation into the role of PDGF, PDGFR in stellate cell activation. This is beyond the scope of what we intended to study and our expertise. We apologise that we are unable to address this important comment by reviewer 1, but sincerely hope that the reviewer understands our limitations.
- The authors have assessed hepatocytes senescence in liver biopsies by p16 staining, but then used p21 in HepG2 in vitro model. Although both can be considered markers of senescence, the choice of either marker should be explained.
We did explain the reason for this in our previous revision. We chose p16 over p21 as p16 has been shown to be associated with established senescence compared to p21. However, we could not find an appropriate p16 antibody for immunofluorescence staining. All p16 antibodies that we tried led to substantial background transillumination , which made it difficult to assess. However, such background transillumination did not occur with p21 immunofluorescence. Hence, we used p16 in immunohistochemistry and p21 in immunofluorescence experiments.
Minor:
Fig4. it is not clear for which of the treatment group (6hs or 24hs) the significance is reached. Please put the asterisk above the bar for which significance is reached
We have addressed this in the figure.

Reviewer 2 Report
Comments addressed.
Author Response
REVIEWER 2
Comments and Suggestions for Authors
Comments addressed.
We thank reviewer 2 for accepting our corrections.
This manuscript is a resubmission of an earlier submission. The following is a list of the peer review reports and author responses from that submission.
Round 1
Reviewer 1 Report
- Please provide more details on human HSC isolation protocol and passage number at seeding.
- The number of experimental replicates for each experiment is not stated.
- It is not clear how the aSMA staining was analyzed. Fig 1C displays aSMA (%), is this a measure of the area fraction occupied by aSMA+ve cells?
- Fig. 2: the low magnification and resolution do not allow for clear identification of p16+ve cells (panel A). Please add an inset with higher magnification for clarity. Moreover, Panel A seems to show lower p16 expression and lower PSR staining than panel B, opposite of what reported in the legend. aSMA expression does not appear to be substantially different in the two panels. Images in Fig.2 seems to have different magnifications, please provide an appropriate scale bar and actual magnification used. Are PSR sections matched (same biopsy) to p16 ad aSMA ones?
- Fig.3, p21 image was already published in “Aravinthan, A., et al., The senescent hepatocyte gene signature in chronic liver disease. Exp Gerontol, 2014. 60: p. 37-45. https://doi.org/10.1016/j.exger.2014.09.011.
- Fig. 3. Images appear to be of different magnification and scale bars are missing. To appreciate the actual extension of p21 and HMGI-C induction, a lower magnification (same as control) should be used. The actual quantification of p21 expression needs to be shown by WB, since it is quite unrealistic that p21 is completely absent in control HepG2 cells. Please provide separate channel images (not overlayed with DAPI) as well.
- p53 is reported in Fig.3 legend but not shown.
- Fig.4: Type of error bars (i.e. SEM, SD) is not reported. This graph shows a three groups comparison (control, 6h, 24h) that was analyzed by Student’s t-test (or Mann-Whitney test), as reported in the Methods section, which is not appropriate. Besides, it is not clear for which of the treatment group (6hs or 24hs) the significance is reached. Please perform a statistic test appropriate for three group comparison and for the type of distribution of the data.
- Fig. 5. Please state number of replicate experiments and define error bars. In a previous publication on the same H2O2-treated hepG2 HepG2 model (Aravinthan, A., et al., The senescent hepatocyte gene signature in chronic liver disease. Exp Gerontol, 2014. 60: p. 37-45), the authors have shown that senescent HepG2 express very high levels of TGFb, LTBP1 and AREG, among other cytokines. How do those published results reconcile with the absence of modulation shown here for TGF1, LTBP1 and AREG?
In the discussion, the authors suggest that incubation of hHSC with conditioned media from senescent HepG2 could potentially stimulate the initiation phase of HSC activation. However, this is a strong over interpretation of the results: they do not state the passage of isolated HSC, which are well known to be subject to culture-induced activation (on plastic or rigid substrates) starting immediately after seeding. Moreover, no data on HSC proliferation, migration, morphology are provided, all of which are important to assess the degree of HSC activation.
A noticeable part of the discussion is dedicated to the anti-fibrotic effect of Sirolimus, whom relevance to the present work is undefined. mTOR inhibitors elicit a complex cascade of effects, from kinase activity inhibition, metabolic changes, inhibition of proliferation just to name a few. Besides, Sirolimus was not used in this work to block SAPS -induced HSC activation by senescent HepG2.
Finally, they propose that aSMA could be used as an early biomarker for liver fibrosis, which is not really a new finding considering that this marker has been used to assess HSC activation since the mid ‘90s.
Minor
- Matherial and Methods: Inconsistent Capital letter use in paragraph titles (lines 70, 96, 130) please correct
Reviewer 2 Report
This study showed that hepatocyte senescence activates the hepatic stellate cell which is a key cell in chronic liver disease including lung fibrosis. This manuscript was well-written and used appropriate methods. However, there are major concerns should be modified be modified before publication.
Major concerns
- Damage to hepatocyte can result apoptosis, necrosis, and senescence. The thesis of ‘senescence can result the human hepatic stellate cells (HSCs) activation’ was recently introduced, however, there are several studies reported that damage to hepatocyte secretes the growth factors, including TGFβ1 or PDGF. What is the difference with previous studies? (What is the novelty of this study?). Also, author confirmed that H2O2 induced only senescence in HepG2 cells, not apoptosis or necrosis? Also, please describe why used H2O2 to induce senescence (not fatty acid or others)
- In this study, author isolated HSCs from human tissue and used it to prove the activation by conditional medium from hepatocyte. Commonly, isolation of HSCs is more complicated process than isolation of hepatocyte and hepatocyte can be isolated during the process of isolation of HSCs. If author used primary hepatocyte than HepG2, which is hepatocarcinoma cells, this study would be more reliable. Is there any reason why author used HepG2 cells?
Minor concerns
- In figure 2, it is difficult to understand the relationship between p16 and αSMA. To understand easily, I suggest to used co-staining method in identical tissue. Also, arrows should be used to indicate the relationship.
- In figure 3, is it same magnification? It looks like magnification is different in control cells and H2O2 treated cells. Please indicate each magnification.
- In table 1, only GAPDH was written in bold. Also, it is better to used italic font in gene name.
Reviewer 3 Report
The authors indicated that hepatocyte senescence is involved in the activation of hepatic stellate cells (HSCs) followed by liver fibrosis progression. This study makes interesting suggestions about the role of senescence hepatocyte observed in liver fibrosis, but there are several points to be noted.
<Introduction>
The description of the factors used in this study should be explained in the introduction. It will helpful to readers.
<Results>
section 3.1.
・In Figure 2, panels A and B seem to be reversed.
・Add the scale bar to the Figure 2.
・The authors should add a description of the PSR.
Section 3.2.
・The authors confirmed senescence of HepG2 cells with fluorescent immunostaining for p21, however the liver biopsy specimens with p16. The authors should explain why they used different markers.
・After induction of senescence in HepG2 cells, the cells were incubated with fresh culture medium for 3 days in Materials and methods (line 100), but 5 days in Result (line 172). The authors should correct it.
・In the legend of Figure 3, it is written that the gene expression of p53, p21, and p16 increased. Please include those data in Figure 3.
・Add the scale bar to Figure 3.
Minor points
・Abbreviations should be defined in parentheses the first time in the paper (e.g., PSR).
・Lowercase and uppercase letters are mixed in section titles. The authors should unify them.
Round 2
Reviewer 2 Report
All issued concerns were solved.
Reviewer 3 Report
Result in the text cannot guarantee credibility unless the experimental results are included. If Figure 3 cannot be shown, it is difficult to discuss Figure 4 as well.